# Recall, understanding, use, and impact of front-of-package warning labels on ultra-processed foods: A qualitative study with mothers of preschool children in Peru

Francisco Diez-Canseco[1]*, Lizzete Najarro[1], Victoria Cavero[1], Lorena Saavedra-Garcia[1], Lindsey Smith Taillie[2,3], Francesca R. Dillman Carpentier[4], J. Jaime Miranda[1,5]

1 CRONICAS Centre of Excellence in Chronic Diseases, Universidad Peruana Cayetano Heredia, Lima, Peru, 2 Carolina Population Center, University of North Carolina at Chapel Hill, Chapel Hill, North Carolina, United States of America, 3 Department of Nutrition, Gillings School of Global Public Health, University of North Carolina at Chapel Hill, Chapel Hill, North Carolina, United States of America, 4 Hussman School of Journalism and Media, University of North Carolina at Chapel Hill, Chapel Hill, North Carolina, United States of America, 5 Department of Medicine, School of Medicine, Universidad Peruana Cayetano Heredia, Lima, Peru

* francisco.diez.canseco.m@upch.pe

**Data Availability Statement:** The focus group transcriptions in Spanish cannot be made publicly

## Abstract

In its efforts to reduce increasing rates of obesity and nutrition-related noncommunicable diseases, Peru implemented front-of-package (FOP) warning labels (also called warnings) on processed and ultra-processed foods in June 2019. The goal was to inform consumers about high levels of sugars, saturated fats, sodium, and trans fats in packaged products. We designed a qualitative study to reveal the recall, understanding, and use of the warnings and to explore the perceived changes in purchasing behaviors among mothers of preschool children in Peru. In mid-2021 we conducted 18 focus groups with 98 mothers from 2 socioeconomic strata (SES) residing in 4 geographically and culturally diverse cities. We analyzed the focus group transcripts for themes relating to the mothers' awareness, understanding, and use of the warnings and the warnings' influence. Our results show that most mothers remembered the warnings and understood their general purpose and meaning, although some warnings were more easily remembered and understood than others. For example, sugar was easier than trans fats. Many mothers considered the warnings in their purchase decisions, although the impact of the warnings on final purchase decisions varied. The warnings were less effective for products that mothers already knew were high in critical nutrients, that they considered essential for certain preparations (e.g., butter), or that they enjoyed (e.g., chocolate). Most mothers reported changes in their purchasing and eating habits due to the warnings, mainly in reducing the frequency and quantity of some processed food intake and opting instead for homemade preparations or warning-free packaged products. Our study shows the usefulness of the FOP warning labels for informing consumers from different settings and SES about the healthfulness of packaged products. The study identifies key areas in which Peru can improve the policy and offers valuable lessons for other countries interested in the implementation of FOP warning labels.

available due to confidentiality agreements outlined in the informed consent form signed by the participants, which specified that "the information you provide will be kept confidential and will be used only for the purpose of the study." However, in accordance with the PLOS Data Policy for qualitative studies, excerpts from the transcripts relevant to this publication and a summary of the group discussions on each topic included in the manuscript (both in Spanish) can be requested from the Universidad Peruana Cayetano Heredia's Institutional Research Ethics Committee. Requests can be directed to Silvana Díaz Mancilla, Manager of ORVEI, via email at duari.orvei@oficinas-upch.pe, phone at +511 3190000 (Extension: 201355), or through the website https://investigacion.cayetano.edu.pe/duari/orvei/. Additionally, the manuscript includes several participant quotations that support the study's findings, adhering to the same policy guidelines.

**Funding:** This study was funded by Bloomberg Philanthropies. Prime awards (N° 46129 and N° 2019-71181) were granted to Barry Popkin at the University of North Carolina (UNC), and subawards (N° 5112886 and N° 5124181) were granted to JM at the CRONICAS Centre of Excellence in Chronic Diseases. The funders had no role in study design, data collection and analysis, decision to publish, or preparation of the manuscript.

**Competing interests:** The authors have declared that no competing interests exist.

## Introduction

In response to rapid increases in obesity and noncommunicable diseases (NCDs) [1], many low- and middle-income countries have adopted policies aimed at reducing consumption of processed and ultra-processed foods high in nutrients of concern [2]. One policy recently implemented in various Latin American countries is the use of front-of-package (FOP) warning labels (also called warnings) on packaged products [3] to inform consumers about potentially unhealthy contents, such as excessive calories, sugars, sodium, or fats, thus motivating healthier, well-informed purchase decisions.

Peru, like many other Latin American countries, has faced increasing mortality from nutrition-related NCDs [4]. In response, in 2013 Peru enacted the Law for the Promotion of Healthy Eating for Children and Adolescents (Law Number 30021) [5], which included FOP warning labels. Multiple barriers and great resistance from the food industry and its political allies delayed the law's implementation for several years. Yet the law's supporters overcame these challenges, and the warnings were implemented in June 2019 [6, 7]. Following Chile's successful experience [8], Peru introduced four black octagonal labels that indicate a processed food or beverage is "high in" sugar, saturated fats, or sodium or contains trans fats according to parameters the Ministry of Health established (Fig 1) [9].

To allow industry time to adapt, the warnings were implemented in two phases, in June 2019 and September 2021, with stricter nutritional thresholds introduced in the second phase [10].

Products that exceed the government-defined limits in sugar, saturated fats, and/or sodium must have a black octagon declaring "high in" each excessive nutrient with the subheading "avoid its excessive consumption." Products with trans fats must have a black octagon declaring "contains trans fats" with the subheading "avoid its consumption" (see Fig 1). Furthermore, any product that qualifies for a warning must include that warning in all of its marketing content (e.g., television, radio, online stores, advertising panels, supermarket inserts, etc.). The Peruvian regulation also bans the sale of products with warnings in schools [11].

Understanding consumers' real-world experiences with warnings, that is, whether and how they notice, understand, and use them, is critical to evaluating policy effectiveness and informing policy development. To date, however, few scientific publications report consumers' experiences using the warnings on foods [12, 13], and no research reports on Peru's warnings specifically.

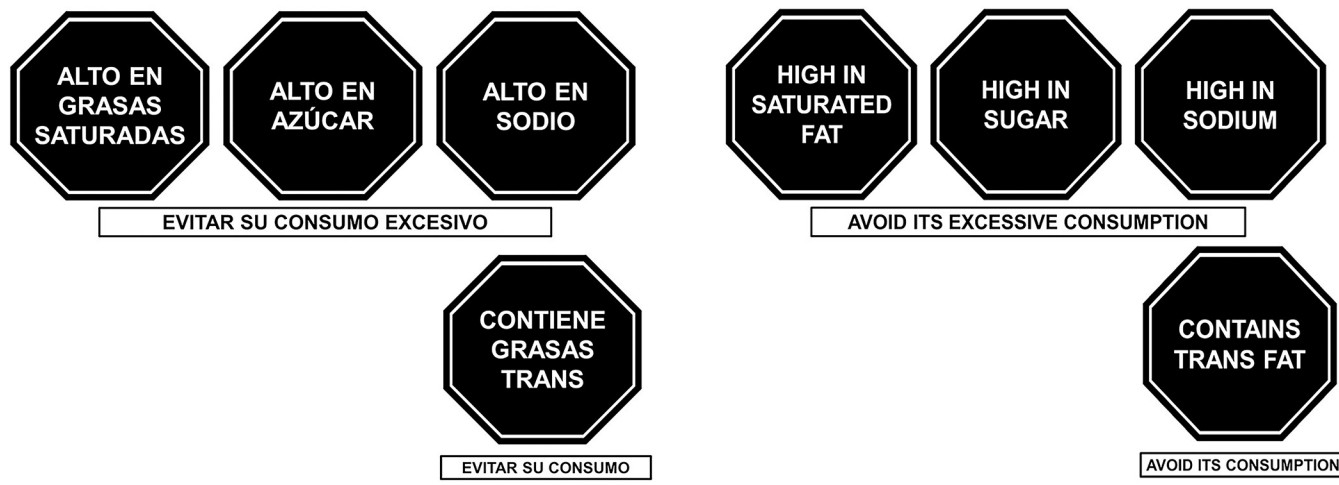

**Fig 1. Peruvian FOP warning labels [9] in Spanish (original) and English.**

In this context, we designed a qualitative study to understand the experiences of mothers of preschool children with the recently implemented warnings, using focus groups of women representing geographically and culturally diverse Peruvian cities and two socioeconomic strata (SES). Our main reasons for working with mothers of preschool children were that eating patterns established at an early age are usually maintained during adulthood [14, 15] and that women are primarily responsible for feeding their children and defining what they eat.

The goals of this research are to discover how mothers remember, understand, and use the warnings and to determine mothers' perceptions of how their purchasing behaviors have changed. With these findings we can identify specific ways to improve the current implementation of warnings in Peru and present valuable lessons for other countries developing similar policies in their own contexts.

## Materials and methods

### Study design

We conducted a qualitative study using focus groups in mid-2021, two years after Peru introduced FOP labels.

### Settings and participants

The participants were mothers of preschool children (3–5 years old) from Lima (a coastal megacity and Peru's capital with a population of 10 million people), Piura (a midsize coastal city of 575,000 people), Ayacucho (a small city in the southern highlands of 255,000 people), and Tarapoto (a small city in the Amazon region of 175,000 people) [16]. We chose these four cities, all capitals of their regions, to reflect the geographic, social, cultural, and nutritional diversity of Peru.

We selected women from two SES in a five-strata classification, labeled from A to E (being A the highest and E the lowest), a categorization used by the Peru National Household Survey. We aimed to conduct four focus groups per city, two with mothers of medium-high SES, SES B, and two with mothers of medium-low SES, SES D. These two SES represent 50% of urban Peruvian households. We defined a participant's SES based on the following variables included in a preliminary survey: (i) head of household's education level; (ii) type of health insurance; and (iii) the house's materials (i.e., floor and walls), the household's services (e.g., internet), and the household's assets (e.g., a vehicle) [17].

We used a purposive sampling method [18] to select participants who matched the following criteria:

1. Is a mother of a child three to five years old

2. lives in one of the four selected cities

3. belongs to SES B or SES D according to the preliminary survey

4. has access to a device with internet connection

5. is able to provide informed consent.

Our aim was to have 5–6 women per focus group or 20–25 women per city and 80–100 women in total.

### Data collection tools

Due to Peru's restrictions associated with the COVID-19 pandemic, the research team held virtual focus groups. The research team developed a focus group guide based on the study

**Table 1. Focus group guide topics.**

| |
|---|
| 1. Feeding a young child: types of foods and decision-making processes |
| 2. Recall and understanding the FOP warning labels |
| 3. Use the FOP warning labels before and during the pandemic |
| 4. Purchase and consumption changes due to the FOP warning labels' implementation |
| 5. Perception and judgment of the FOP warning label policy |

objectives. The guide was designed to cover 5 key topics related to food-decision making as well as the pathway from FOP warning label exposure to behavior change (e.g. recall, understanding, use, effect on purchases) [19] (see Table 1).

We used three images to facilitate the conversation at different moments of the focus group: (1) a picture of various ultra-processed foods and beverages (e.g. dairy products, breakfast cereals, sugar-sweetened beverages, etc.) to show the variety of packaged products that we would be discussing (2) a black octagon without text for participants to say which of the different texts they remembered (e.g. 'high in sugar'), and (3) all four Peruvian octagons, to remind them the current content of the warnings and start discussing about their meaning.

Additionally, we used a sociodemographic form to collect age, education level, work status, children's demographics, and place of purchasing packaged food.

## Procedures

We piloted the focus group questions and the logistics of the virtual sessions with three groups in Lima and do not include their content in the current analysis. We formed a recruitment team in each city with a supervisor who verified the inclusion criteria of each woman before inviting her to a focus group. Potential participants were identified using purposive sampling, with initial contact being made in person or by telephone. A total of 153 women were invited, of whom 98 attended the focus groups.

The recruitment was performed between July 2nd and August 3rd 2021, and the focus groups between July 8th and August 5th 2021. All focus groups met virtually using Zoom, and we recorded the audio. The average duration was 94 minutes, ranging from 66 to 119 minutes.

An experienced female social scientist, with 15 years of experience in qualitative data collection and analysis in Peru, facilitated the groups and recorded each participant's consent, answered questions about their participation, and managed the group discussion. A nutritionist assisted the facilitator, took notes, and addressed mothers' nutrition-related questions after the session.

## Data analysis

We transcribed the audio recordings verbatim and used ATLAS.ti 9 software to conduct a content analysis [20]. Two researchers, including the facilitator of all the groups, coded the content. The research team developed an initial codebook for categorizing and describing themes based on the questions in the focus group guidelines plus relevant topics that arose after reading two randomly selected transcriptions (10%). Throughout the analysis researchers discussed unanticipated emerging themes to decide whether to add new codes to the codebook. The research team met weekly to discuss findings and the potential inclusion of new codes.

After we completed the coding, we used a coding matrix in Excel [21] to summarize the codes. We present results related to topics 2, 3, and 4 of the focus group guide (see Table 2).

## Ethical considerations

The Institutional Review Board at Universidad Peruana Cayetano Heredia approved the study on January 27th 2021 (Approval 681-36-20). All participants gave their informed consent. Due

**Table 2. Participants' characteristics.**

| Variables | SES B (n = 45) | SES D (n = 53) | Total (n = 98) |
|---|---|---|---|
| Mother age* | 34.0 (20–49) | 30.6 (20–49) | 32.0 (20–49) |
| Education Level | | | |
| *Did not finished school* | 0% | 5% | 3% |
| *Completed secondary school* | 9% | 51% | 32% |
| *Uncomplete superior education* | 9% | 25% | 17% |
| *Completed superior education* | 75% | 19% | 45% |
| *Post-graduate* | 7% | 0% | 3% |
| Has paid work | 73.3% | 39.6% | 55.0% |
| Number of children* | 1.8 (1–4) | 2.1 (1–5) | 1.9 (1–5) |
| Children age* | 3.9 (3–5) | 3.8 (3–5) | 3.9 (3–5) |
| Children female gender | 60.0% | 45.3% | 52.0% |
| Place of purchase of packaged food | | | |
| Supermarket | 62.2% | 3.8% | 30.6% |
| Local market | 37.8% | 96.2% | 69.4% |

*Mean (range).

to the COVID-19 pandemic restrictions, we shared a digital informed consent form and then reviewed it during a phone call. Those who agreed to participate gave oral consent during the call, and their consent was audio recorded.

To compensate for their time, all participants received a voucher of approximately USD $10 for a supermarket. Those who needed it also received a telephone recharge to allow them to have internet and stay connected during the virtual group.

## Results

### Participants' characteristics

We conducted 18 focus groups instead of the original 16, because attendance was lower than expected at 2 meetings. A total of 98 women participated (46% from SES B, 54% from SES D), with 22–29 participants per city and on average 5.5 people per session. Table 2 describes some essential characteristics of the participants and their children by strata.

### Recall of the warnings

Nearly all of the participants easily identified the black octagons as warnings. Most women remembered seeing the warnings for the first time in 2019 or 2020, before the beginning of the COVID-19 pandemic.

Most women also remembered seeing the warnings on packaged foods. Some mentioned the warnings in reference to processed products, and others mentioned specific foods or beverages, for example, cookies or other sweets, salty snacks, sugary drinks, chocolate milk, cold cuts, breakfast cereals, or yogurts. Additionally, the vast majority of mothers remembered seeing the warnings in television advertising of processed products, followed by billboard ads, social networks, and supermarket inserts.

> *[The warnings]. . .are mostly in everything that is packaged, canned. . . in biscuits, in dairy products. In reality, right now I think most of the products contain this type of symbol, where we can identify some ingredients such as high in sugar, saturated fats. . . .*
> *(Piura_SESB_FG04)*

When invited to remember the messages in the warnings, the mothers frequently mentioned "high in sugar" and "high in (saturated) fats," followed by "high in sodium." "Contains trans fats" was barely mentioned. When asked about the advice accompanying the octagons ("avoid excessive consumption" and "avoid consumption"), only some remembered them quickly and accurately.

Only seven women, mainly from Ayacucho, said they had never seen the octagons before. These participants explained that they paid very little attention to package labels or almost never bought packaged foods.

## Understanding of the warnings

As expected, the mothers understood that the presence of more octagons represented a stronger warning and the idea that the product is more harmful for their health than products with fewer or no warnings. Additionally, the "high in" message was clear for all mothers, who understood that the labeled product had "too much" of a nutrient.

> ["High in" means] that it's in too much quantity and that, if we consume too much... it's going to damage our health. (Ayacucho_SESB_FG02)

They had no doubts about the "high in sugar" octagon's meaning. In fact many women spontaneously associated the warnings with the "high in sugar" or "high in (saturated) fats" messages. Yet the word "sugar" was much more common and better understood than "saturated fats" and "sodium." Participants from Ayacucho and Tarapoto, the smallest cities, and those from the lower SES D particularly noted this lack of clarity.

Many mothers could not clearly say what sodium and saturated fats were but nevertheless had a clear understanding that both were harmful and commonly linked to specific diseases. They associated sodium with high blood pressure and heart and kidney diseases and saturated fats with fried products, oily products, and "bad" fats (in contrast to "healthy" fats), all considered negative for health.

> The sodium affects the organism, in vascular diseases, your kidneys, your heart, I don't know, it is... it produces a series of diseases... if you consume it excessively.... (Tarapoto_SESB_FG03)

> Saturated fat is a bad type of fat, because there is also good fat, like in avocado for example. So, it is a bad type of fat, which affects the circulatory system and... other organs of our body. (Lima_SESB_FG02)

The trans fats warning was consistently the least understood and remembered. Most mothers across all cities and SES said that before the focus group they had not seen this warning or did not know what it meant. Only very few participants from the higher SES had a clearer idea. Additionally, very few of the mothers remembered that the trans fats warning recommended avoiding consumption. When this message was shown during the conversation, various mothers shared their concerns, and some reflected on why the authorities allow production of foods that are harmful, such as those containing trans fats.

> I have never seen or realized where the information about fats, trans fats was.... Now we should be more observant, I think, no? To be able to see what that octagon says... and to be able to know what food we should avoid. Now, I don't understand, why do they produce [this food] if it is harmful, and it is a food that we should not eat? No? In other words, they produce it but then tell us "Don't eat this." So... what is the idea? (Lima_SESD_FG01)

Along with these concerns, many mothers expressed the necessity and expectation for receiving more information. Some suggested that, in addition to the octagons, both the packaging and the advertising of processed products should include information about the health consequences of their excessive consumption.

## Use of the warnings before and during the pandemic

Mothers frequently reported using the warnings when making purchase decisions, but the value and attention given to the warnings varied based on the circumstances and the type of processed and ultra-processed foods being considered. For example, some mothers did not take warnings into account when considering products they already knew were high in sugar or fat, products they were very used to purchasing, or products they really liked. On the contrary, several mothers shared their surprise when they realized that some products they regularly consumed and considered healthy were potentially harmful. The surprise was greater—expressed as a feeling of being scammed—when those products were targeted to children and advertised as healthy and nutritious, as were some yogurts and milks.

*It is something that impacts you, if after having consumed [a product] so much and for many years, they warn you that it can harm you, that it has a lot of sugar, after you have consumed it so much. . . . Some pediatricians tell you that you can give your children milk or yogurt for constipation. . . but they should warn you that it has a lot of sugar. (Tarapoto_SESB_FG02)*

*I saw the octagon in a milk. . . there cannot be that [an octagon] in milk! For me the milk is essential for children, no? I felt cheated. (Lima_SESD_FG03)*

Some mothers also shared concern and fear over the potential harmful effects of long-term consumption of labeled foods on children's health. Others expressed feelings of guilt after realizing they had been buying unhealthy products.

*In my case, I felt super weird. . . . As if I was getting sick myself, no? I mean, with my own hands, my own choices. In this way, this information is beneficial for us, being able to know. . . . So, you can reduce the consumption, you have more information about what the product offers. . . . But at the beginning [I thought] "I am killing myself." Now it is changing, I am improving my choices when buying and deciding what foods to consume. (Tarapoto_SESD_FG01)*

In households where someone was sick, the warnings were especially useful when making purchase decisions.

*In my house lives my grandmother, who is diabetic and she eats practically only natural food. . . . And if I see, on a package, that there is an octagon. . . I need to think if I buy it or not. . . . I think I would not [buy it], I mean, it is harmful, no? And there I need. . . to take a decision. (Lima, SESD, FG03)*

The mothers did not clearly consider any one of the four warnings more important than the others or use one more commonly when buying packaged foods. Some said they gave more attention to warnings related to the health conditions of their children or relatives, for example, avoiding "high in sugar" products if somebody had diabetes or was overweight and avoiding "high in sodium" products if someone was hypertensive. Additionally, based on the information they received during the focus groups, some mothers considered the trans fats

label the most worrying. On the one hand, they worried because they did not know what trans fats were. On the other hand, they worried because of the message that accompanies the warning recommending avoiding its consumption.

A few mothers also used the warnings to talk to their children about the products' contents to dissuade them from consuming labeled products.

*Sometimes my husband buys potato chips, a soda, and there are the octagons, no? I tell my daughter "This soda has too much sugar," I told her, "In the long-term, it will harm us if we consume it too much." (Piura_SESB_FG02)*

In some cases the absence of an octagon on an ultra-processed food or beverage was used as an argument to justify the product as healthy.

*I don't drink soda, but in birthday celebrations they always invite you a soda, right? So, he told me "Drink, because Guarana soda doesn't have octagons, it is super natural.". . . He even shows me the bottle, the big 3-liter bottle and tells me "Look, there's no octagon, right? Do not worry." I was surprised because normally all sodas have an octagon and this soda did not. (Piura_SESB_FG02)*

Most mothers agreed that the impact of the warnings was stronger when they were first implemented. At that time it was common to talk about the warnings and to carefully read the nutritional information on packaged products. Similarly, participants said they were more aware of the presence of warnings during the first months of implementation. They still considered warnings useful, but when we conducted the focus groups they had already become accustomed to the warnings' presence and had changed some purchasing and eating habits.

Regarding the use of warnings during the COVID-19 pandemic, responses were not conclusive. The pandemic made some participants more conscious of the importance of their diets, especially if they or a relative were at risk of a severe disease. These participants reported being more aware of the warnings when selecting packaged foods.

*Now I use them [the octagons] more, I observe them more because. . . this pandemic has shown us that we should eat well, because sometimes, we mess up our eating and eat anything like junk food and we're not getting fed. Many people have been affected by the pandemic because their diet is poor, we do not eat as it should be. (Lima_SESD_FG01)*

Others increased their purchases of packaged products during the pandemic because of the uncertainty accompanying lockdowns and travel restrictions. Finally, some said they shopped faster than usual due to the fear of contagion, giving less attention to product information, including the warnings.

## Purchase and consumption changes after the warnings' implementation

Many mothers across all cities and both SES said that after the warnings appeared, they continued consuming many of the processed products they used to consume but in less quantity and/or less frequency. A usual justification for purchasing labeled products was that their intake was not excessive nor frequent.

*It is not that I never consume them [products with octagons], I keep consuming them even though [the octagons] are there. For example, the instant soups. . . . To make me feel good, I say "No. . . I don't consume it in excess, and it won't hurt me," that is what I think. So, it is*

*not that I consume it daily, but once every 2 weeks, I try to prepare it because it is delicious. . ..
I have cravings and I consume it, but not in excess. I think it [the octagon] moderates the consumption [of those products]. (Ayacucho_SESB_FG04)*

Some mothers considered intake of some labeled ultra-processed products, especially sugary foods, occasional indulgences or treats for themselves or their children.

*I also consume packaged products from the supermarket, for example butter, olive oil, also
yogurt, cookies for my child, crackers or vanilla cookies, and an occasional chocolate as a treat
for my child, no? However, it is not very often, but yes, yes, we consume those products.
(Piura_SESB_FG02)*

Some mothers mentioned that the warnings prompted them to stop buying some of the processed foods they bought previously.

*At home, my mom used to buy Pulp [a packaged fruit nectar], we drank it for breakfast, with
bread and ham. But with all the octagons, we crossed it off from our shopping list.
(Tarapoto_SESD_FG01)*

In some cases they replaced the labeled products with less processed or healthier alternatives, including foods made at home. For example, raisins or dried fruits replaced sweets, homemade juices replaced packaged fruit nectars, oats replaced chocolate milk, or homemade popcorn replaced packaged popcorn. Other mothers reported replacing one processed product with a similar processed product without warnings—a replacement that was not necessarily healthy.

*Some products that we used to consume frequently, we have tried to replace them, for example, [packaged] chocolate milk. . . now I prefer to make it with chocolate with milk. . . using healthier products. . . or replace it, for example, with* chufla *[a traditional homemade dessert], or with oat, right? Because in the long run it has consequences both for us, as adults, and for children. . . now we take better care of health, it depends on us. (Piura_SESB_FG04)*

When octagons appeared, I was really surprised because Cheetos didn't have octagons.
Since then, I stopped eating Lays potato chips and Chizitos [which had octagons] and preferred Cheetos because they did not have octagons. (Ayacucho_SESB_FG04)

*Regarding sausages. . ., some I found had octagons, but then I found one without octagons. So,
I'd better take the one without octagons, right?" (Lima_SESB_FG02)*

Adults of the same household sometimes differed in their eating habits and preferences, which had been shaped in their early years. These differences might lead to opposing views regarding whether children should avoid foods with warnings.

*In my opinion, when an octagon appears, it [the product] is not good, because it has a lot of
fat, because it has a lot of salt. I am the one who limits the consumption. Because my husband
says "No, give it to her [my daughter]. . . my parents gave a mountain of cookies and nothing
happened to me." But I have grown up in a different environment, my mother used to say
"vitamins are in the pot, in what you cook." I think that, if you consume too much soda, cookies, Lay's potato chips, which come with a lot of salt, sugar, in the long term that harms our*

*kidneys, our liver, because those are the organs that synthesize those minerals. I always try to take care of them. (Tarapoto_SESB_FG02)*

Some mothers stated that they had continued consuming some processed products in a similar amount or frequency regardless of the number of warnings. This usually happened with products they thought were good for children (e.g., yogurt), that their children really liked, that they themselves enjoyed, or that they considered "essential" for cooking (e.g., butter) or making specific preparations (e.g., mayonnaise).

*Although they have 50 octagons, I have not stopped consuming chocolates. It is the only thing I eat, for sure, knowing that it has octagons. . . . Other products with octagons I prefer not to buy them. If I see the octagon, I prefer to look for an alternative that does not have octagons or that is healthier. (Lima_SESB_FG02)*

*When I see that the product says "high fat" but I need to use it, I do use it. The truth, I do use it, as a treat. . . . For example, I do use mayonnaise with chicken and rice [traditional Peruvian dish], even though it is bad for us, but we are used to it. (Piura_SESD_FG03)*

Finally, a few participants explained their lack of change saying that they usually do not pay attention to the information on the package.

## Discussion

This qualitative study aimed to describe the recall, understanding, and use of the FOP warning labels two years after their implementation in Peru and to explore warnings-related changes in purchasing behaviors among preschool children's mothers from different cities and SES. Our results show several similarities and some differences in how mothers with different backgrounds remember, understand, and use nutritional warnings. Two years after the warnings' implementation, most mothers clearly recognized them and knew their purpose. However, warnings for sugar were more easily understood than those for sodium and saturated fats, while the trans fats warning was the least understood and remembered. In addition, mothers from larger cities and a higher SES appeared to have a better understanding than those from smaller cities or a lower SES. Importantly, the women commonly noted warnings when making food purchases and reported that this led to a reduction in the quantity and/or frequency of their consumption of processed foods. However, we found some heterogeneity. For example, the mothers used warnings less with products they considered essential for some preparations.

Our findings corroborate evidence from experimental studies on the usability and comprehensibility of FOP warning labels. According to a systematic review, FOP nutritional warnings like the Peruvian octagons are easy to understand, help consumers identify "high-in" products, and discourage their purchases of those products [19]. Our study found lower recall and understanding of some warnings compared to others, which might have various explanations. First, some nutrients are more familiar to Peruvian mothers than others (sugar is better known than trans fats). In that sense, this difference has more to do with the nutrient itself than the label. Second, the trans fats warning appears much less frequently on packaging than the other three warnings. Third, Peru did not conduct an information campaign to support the warnings' implementation. In fact some mothers demanded more and better information regarding the warnings. Beyond this, most women from all cities and SES understand the octagons' meaning and know that the more octagons, the less healthy the product. Moreover, even

though some mothers did not know what a specific nutrient was, the warning still helped them know that a product was unhealthy.

Our results also show that mothers were less likely to use the warnings for products they considered to be essential for cooking, such as culinary ingredients like oil or butter, or for products they already considered unhealthy, such as chocolates. Taken together, these results suggest that the warnings' impact is greatest among packaged products where consumer confusion about healthfulness is likely to be high.

Our results show that women in Peru do not use the warnings in the same way for all products. Similarly, a qualitative study with mothers in Chile, a year after that country implemented its FOP policy, reveals that Chilean mothers found the octagons more useful with some packaged products than with others and used the labels to guide their children and to talk with them about the products' healthfulness [12]. The perceived importance of the warnings for talking better purchase decisions is also highlighted by the 2021 ENSANUT survey in Mexico, that reported that 60.5% of parents declared that the FOP helped them to select healthier foods for their children [22]. Additionally, results in both Peru and Chile captured mothers' disappointment and feelings that they had been misled when they realized that some products they previously considered healthy now contained warnings [12].

Our participants commonly reported reducing the amount and frequency of purchases of packaged products with warnings. This self-reported behavioral change is in line with sales data that the market research company Kantar collected in Peru in September 2019. Those data revealed a reduction of 4% in the sales of labeled products in July 2019 from July 2018, with higher decreases in cookies (6%), sugary drinks (7%), and especially yogurts (27%) [23]. This last figure is aligned with the women's narratives about their differential use of octagons and also their discomfort at seeing warnings on products they previously considered healthy, such as yogurts.

The perception of many mothers in our study, that the warnings reduced their purchases of unhealthy products, is similar to Brazil, where a qualitative research conducted before the implementation of their FOP labels [24] showed that most women believed that they would reduce the consumption of products displaying FOP. Additionally, the evaluation of the Chilean food labeling law showed a reduction in purchases of "high in" products [25]. Together these results highlight the influence of FOP warnings to reduce mothers' purchases of unhealthy products.

However, the attention and use of FOP warnings depends on other factors as well. For example, the women in our study reported paying lower attention to octagons when purchasing products they were used to, liked very much, or considered essential, which is also consistent with the qualitative results from Chile [12]. FOP information is used less after brand loyalty is established [26]. In addition, in a set of focus groups conducted after the third phase of the Chilean law, mothers reported that they did not always make purchasing changes in response to the FOP warning labels if they could not afford the healthier option [13]. Together, these results suggest that the warning labels may be more likely to influence purchases of some products more than others, that the influence of labels may change over time, and lastly, to maximize the FOP policies' effect on healthy food purchases, it is also important to ensure that healthy alternatives to products that carry the label are affordable for consumers.

A novel result is the value several mothers placed on the warnings when making purchasing decisions during the COVID-19 pandemic. This finding is important, because it supports the utility of the warning labels for making healthier decisions when choosing a food or beverage, a procedure to which consumers often invest less than 10 seconds [27]. A clear and simple message regarding the healthfulness of processed foods is valuable in Peru, where one in four packaged products has no nutritional information [28].

## Policy implications

Our results show the usefulness of the FOP warning labels policy in informing consumers about the nutritional content of packaged products, including women in lower SES and smaller urban situations. First, the findings reveal that individual labels for each nutrient helps mothers understand that unhealthier products have more warnings. Second, even when mothers do not fully understand the meanings of some nutrients, the warnings help them identify unhealthy products.

The warnings are frequently understood and commonly used across different settings and SES, and mothers report changes in food purchasing. In addition the study findings offer valuable information for improving policy implementation in Peru and lessons for other countries. First, communication campaigns are essential before and during the policy implementation. The campaign strategies and contents should be tailored to different audiences but also should tackle a possible reduction in the population's interest in the warnings after a few months or years. Second, although the available evidence in Peru shows a reduction in the amount of sugar and fats in the best-selling processed products after the nutritional warnings' implementation [29], complementary measures are key to improving nutrition at the population level, including actions for increasing the availability and accessibility of natural and minimally processed foods. After seven years of FOP labels in Chile, the population seems to be experiencing a "labels fatigue" due to the prevalence of warnings, which could reduce the policy's effectiveness [13]. Third, the absence of a warning on a product's package could give the false impression that it is healthy. The latter is particularly relevant in Peru, since processed products in small packages (i.e., less than 50 square centimeters) are exempt from carrying the warnings, even when most of those products exceed the limits of sugar or saturated fats established in the regulation [30].

## Strengths and limitations

An essential strength of our study is the diversity of the sample, which made it possible to identify important similarities and some differences in the discourses and experiences of mothers from different SES and regions. It is also important to acknowledge some of the study's limitations. First, the two years that had passed since the initial policy implementation could affect women's recall, but at the same time that allowed us to assess their views after the novelty of the warnings had passed. Second, social desirability, especially in a group conversation, could have biased the responses of some mothers when describing their use of the warnings and changes in their purchasing behaviors. However, we did collect responses from women who stated that they continued to buy processed products as before and some who could not recall ever having seen an octagon. Third, the use of labels as well as the purchase changes reported by the participants could have been influenced by variables that were not considered or recorded in the recruitment, such as having a chronic condition in the household. Fortunately, some of these variables emerged in the groups, and women reported that the presence of diseases such as hypertension and diabetes in their families, affected their perception and use of the warning labels. Being these conditions highly prevalent in Peru, these results reflect the daily experience of the Peruvian population exposed to the labels. Finally, it is easier for participants to be distracted by stimuli around them during online focus groups. However, attendance and participation in the focus groups was almost always good, and when the attendance was low, we organized new groups.

## Conclusion

In Peru most mothers of preschool children from different cities and SES remember and understand the purpose of the FOP warning labels two years after their implementation. Most

mothers commonly used the labels when deciding which products to buy and reported changes in purchasing habits due to the warnings. However, the labels' impacts seem to be heterogenous. The FOP warning labels policy achieves its main objective of informing consumers and helping them make healthier choices, yet further work is needed to strengthen the policy's impact and ensure its effectiveness over time.

## Author Contributions

**Conceptualization:** Francisco Diez-Canseco, Lizzete Najarro, Lorena Saavedra-Garcia, Lindsey Smith Taillie, Francesca R. Dillman Carpentier, J. Jaime Miranda.

**Formal analysis:** Lizzete Najarro, Victoria Cavero.

**Funding acquisition:** Francisco Diez-Canseco.

**Investigation:** Francisco Diez-Canseco, Lizzete Najarro, Victoria Cavero.

**Methodology:** Francisco Diez-Canseco, Lizzete Najarro.

**Project administration:** Francisco Diez-Canseco, Lorena Saavedra-Garcia, Lindsey Smith Taillie, J. Jaime Miranda.

**Supervision:** Francisco Diez-Canseco.

**Writing – original draft:** Francisco Diez-Canseco, Lizzete Najarro.

**Writing – review & editing:** Francisco Diez-Canseco, Lizzete Najarro, Victoria Cavero, Lorena Saavedra-Garcia, Lindsey Smith Taillie, Francesca R. Dillman Carpentier, J. Jaime Miranda.

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
