## [Decision Letter · Decision Letter 0]

15 Aug 2024

PGPH-D-24-01396

Recall, understanding, use, and impact of front-of-package warning labels on ultra-processed foods: A qualitative study with mothers of preschool children in Peru

Dear Dr. Diez-Canseco,

Thank you for submitting your manuscript to PLOS Global Public Health. After careful consideration, we feel that it has merit but does not fully meet PLOS Global Public Health’s publication criteria as it currently stands. Therefore, we invite you to submit a revised version of the manuscript that addresses the points raised during the review process.

Please note that we have only been able to secure a single reviewer to assess your manuscript. We are issuing a decision on your manuscript at this point to prevent further delays in the evaluation of your manuscript. Please be aware that the editor who handles your revised manuscript might find it necessary to invite additional reviewers to assess this work once the revised manuscript is submitted. However, we will aim to proceed on the basis of this single review if possible. 

We look forward to receiving your revised manuscript.

Kind regards,

Avanti Dey, PhD

Staff Editor

Journal Requirements:

Additional Editor Comments (if provided):

Reviewers' comments:

Reviewer's Responses to Questions

**Comments to the Author**

1. Does this manuscript meet PLOS Global Public Health’s publication criteria? Is the manuscript technically sound, and do the data support the conclusions? The manuscript must describe methodologically and ethically rigorous research with conclusions that are appropriately drawn based on the data presented.

Reviewer #1: No

2. Has the statistical analysis been performed appropriately and rigorously?

Reviewer #1: No

3. Have the authors made all data underlying the findings in their manuscript fully available (please refer to the Data Availability Statement at the start of the manuscript PDF file)?

Reviewer #1: Yes

4. Is the manuscript presented in an intelligible fashion and written in standard English?

Reviewer #1: No

5. Review Comments to the Author

Reviewer #1: The article is interesting because it could contribute to Peru's current policy on front-of-package labeling. However, I must highlight some points in the article that need to be adjusted in a major way. I list these points according to the position of the line:

line 76: The phases of implementation of labeling are described to allow the industry to adapt, and the table shows the profiles of each stage. However, this article does not discuss changes between implementation phases or evaluate seals between products. I suggest considering avoiding and/or removing paragraph and table 1.

line 113: It is mentioned that participants were selected from a national household survey. However, the reference of the survey is not detailed or placed, what year is it, who made it, etc.?

It also does not detail how the women were selected, was it random, was it for convenience, for example, what was the total sample and which was the selected one, how were they contacted? what was done in case they were not contacted? What was the positive response rate? etc.

line 117: It states the variables that were used to evaluate SES, could you please state what was the basis or reference of how the SES was constructed for the selection?

line 130: Data collection tools. Table 2 describes the themes used in the focus group guide. However, the methodology does not explain whether any model, framework or theory was used to measure the themes of recall, use, change of purchase, perception or judgment. That is, how these variables were defined in order to develop the guide.

Also, it does not detail how the guide was developed. Was it constructed? or adapted from one already developed and validated?

line 134: The use of images is mentioned, 1) could you explain why these were used, 2) which category of UPP was used and why, 3) what type of beverage, perhaps it would be worthwhile to provide examples of these.

line 147: describe the profile and experience of the person who conducted the focus groups.

line 157: describe what type of analysis was performed: content, thematic, narrative, grounded theory or discourse analysis?

line 177: Table 3 does not describe the mothers' level of education (an important variable for the study) or the sociodemographics of the children as described in lines 136-138.

Line 172: The subtitles of the results do not coincide with the topics described in table 2, for example, topic 5 (table 2) is not mentioned and others seem to have been edited for convenience.

Line 386: Much of the discussion is devoted to discussing a summary of the results. It is compared with a systematic review and a study from Chile, however due to the magnitude of Latin America countries that have implemented FOP warnings there is a need for a comparison at the same level. There are studies from Brazil, Canada and Mexico with a qualitative character that could be discussed.

line 475: I think more can be said about strengths and limitations. For example, the sample was obtained secondarily from a national survey and was not specifically identified for this study, such as education, and regular consumers, but specific inclusion criteria were identified that could have a homogeneous geographic and SES distribution.

Or for example, that although possible they did not remember after two years but were used example images of the products. Or for example, self-reported use or purchase may be influenced by educational level or the presence of previous chronic diseases (the latter was not included), etc.

6. PLOS authors have the option to publish the peer review history of their article (what does this mean?). If published, this will include your full peer review and any attached files.

**Do you want your identity to be public for this peer review?** For information about this choice, including consent withdrawal, please see our Privacy Policy.

Reviewer #1: No

---

## [Decision Letter · Decision Letter 1]

24 Oct 2024

Recall, understanding, use, and impact of front-of-package warning labels on ultra-processed foods: A qualitative study with mothers of preschool children in Peru

PGPH-D-24-01396R1

Dear MPH Diez-Canseco,

We are pleased to inform you that your manuscript 'Recall, understanding, use, and impact of front-of-package warning labels on ultra-processed foods: A qualitative study with mothers of preschool children in Peru' has been provisionally accepted for publication in PLOS Global Public Health.

Best regards,

Hasanain Faisal Ghazi, phd

Academic Editor

Reviewer Comments (if any, and for reference):

Reviewer's Responses to Questions

**Comments to the Author**

1. If the authors have adequately addressed your comments raised in a previous round of review and you feel that this manuscript is now acceptable for publication, you may indicate that here to bypass the “Comments to the Author” section, enter your conflict of interest statement in the “Confidential to Editor” section, and submit your "Accept" recommendation.

Reviewer #2: (No Response)

2. Does this manuscript meet PLOS Global Public Health’s publication criteria? Is the manuscript technically sound, and do the data support the conclusions? The manuscript must describe methodologically and ethically rigorous research with conclusions that are appropriately drawn based on the data presented.

Reviewer #2: Yes

3. Has the statistical analysis been performed appropriately and rigorously?

Reviewer #2: N/A

4. Have the authors made all data underlying the findings in their manuscript fully available (please refer to the Data Availability Statement at the start of the manuscript PDF file)?

Reviewer #2: Yes

5. Is the manuscript presented in an intelligible fashion and written in standard English?

Reviewer #2: Yes

6. Review Comments to the Author

Reviewer #2: This is a very interesting and relevant article and you have fully addressed the previous comments. My only comment is that a few sentences on further research needs are missing.

There are also number of typos that have crept in and another thorough read will be necessary. Here is what I have picked up:

P7 Table 2:

- Mother's age

- Children's age

- did not finish school

- line for postgraduate has slipped

P6, line 144/f: to remind them of the current content of the warnings and start discussing their meaning.

p16, line 442: importance of the warnings for taking better

p19, line 524: As these conditions are highly prevalent in Peru

7. PLOS authors have the option to publish the peer review history of their article (what does this mean?). If published, this will include your full peer review and any attached files.

**Do you want your identity to be public for this peer review?** For information about this choice, including consent withdrawal, please see our Privacy Policy.

Reviewer #2: No
